# Combination of triciribine and p38 MAPK inhibitor PD169316 enhances the differentiation effect on myeloid leukemia cells

**Yuri Sato-Nagaoka[1]°, Susumu Suzuki[1,2,3]°, Souma Suzuki[2], Shinichiro Takahashi[1,2,3] \***

**1** Department of Clinical Laboratory, Tohoku Medical and Pharmaceutical University Hospital, Sendai, Japan,
**2** Division of Laboratory Medicine, Faculty of Medicine, Tohoku Medical and Pharmaceutical University, Sendai, Japan, **3** Institute of Molecular Biomembrane and Glycobiology, Tohoku Medical and Pharmaceutical University, Sendai, Miyagi, Japan

° These authors contributed equally to this work.
\* shintakahashi@tohoku-mpu.ac.jp

**Data Availability Statement:** All relevant data are within the manuscript and its Supporting information files. The gene expression datasets have been deposited in the NCBI Gene Expression

## Abstract

Differentiation therapy with all-trans retinoic acid (ATRA) is well established for acute promyelocytic leukemia (APL). However, the narrow application and tolerance development of ATRA remain to be improved. A number of kinase inhibitors have been reported to induce cell differentiation. In this study, we investigated several combinations of these kinase inhibitors. Recently, we revealed that the Akt inhibitor triciribine (TCN) efficiently induces differentiation of NB4 APL cells and acute myeloid leukemia (AML) M2-derived HL-60 cells through activation of the ERK/MAPK pathway. In the present study, we found that the p38 MAPK inhibitor PD169316 had profoundly enhanced the TCN effect for differentiation of NB4 and HL-60 cells. Morphologically, the combination of these two agents efficiently reduced the nuclear-to-cytoplasmic ratio and induced the expression of myelomonocytic markers (CD11b, CD11c) and some ectopic markers (erythroid glycophorin A, lymphoid CD7 and CD20), as determined by PCR and flow cytometry analyses. Western blotting analysis revealed that these agents efficiently induced phosphorylation of ERK. To clarify the molecular mechanisms involved in the TCN and PD169316-induced differentiation, we performed microarray analyses using NB4 cells. Pathway analysis using DAVID software indicated that "viral protein interaction with cytokine and cytokine receptor" and "cytokine-cytokine receptor interaction" were enriched with high significance. Real-time PCR analysis demonstrated that genes for components of these pathways, including chemokines like *CCL1*, *CCL2*, *CCL3*, *CCL5*, and *CXCL8* as well as cytokines and receptors like *CSF1*, *IL-10*, *IL-10RA*, *IL-10RB*, *IL-1β*, and *TNFSF10*, were upregulated in NB4 and HL-60 cells during TCN and PD169316-induced differentiation.

Omnibus (http://www.ncbi.nlm.nih.gov/geo/) under the GEO series accession number GSE235632.

**Funding:** Grant-in-Aid for Scientific Research (21K07346) from the Ministry of Education, Science and Culture, Japan, to ST. Kyowa-Kirin Research Support (Kyowa-Kirin Co. Ltd.), Japan, to ST. Daiichi-Sankyo Research Support (Daiichi-Sankyo Inc.), Japan, to ST. The funders had no role in study design, data collection and analysis, decision to publish, or preparation of the manuscript.

**Competing interests:** The authors have declared that no competing interests exist.

## Introduction

Acute myeloid leukemia (AML) is a fatal disease characterized by infiltration of bone marrow, peripheral blood, and other tissues by clonal, proliferative, and abnormally differentiated cells of the hematopoietic system [1]. Although standard cytarabine-based chemotherapies can cause severe side effects, this agent is widely used to treat all types of AML. The high intensity of standard chemotherapy protocols followed by hematopoietic stem cell transplantation is unsuitable for many elderly patients, resulting in treatment failure and poor long-term prognosis [2]. Differentiation therapy with all-trans retinoic acid (ATRA) is well established for a specific type of AML, acute promyelocytic leukemia (APL). While the advantages of differentiation therapy with ATRA for APL are the high efficacy and moderate side effects, this therapy has been the only dramatic therapeutic advance for AML in the last three decades [2]. Furthermore, the tolerance development and narrow application of ATRA remain to be improved.

Activation of kinase signal transduction pathways can contribute to leukemogenesis through cell differentiation blockade and aberrant cell proliferation [3]. In addition, kinase inhibitors have been proposed as differentiation agents for AML [4–6]. Consequently, targeting of signaling pathways is considered attractive for molecularly targeted therapy in AML.

A number of kinase inhibitors have been reported to induce cell differentiation, including cyclin-dependent kinase inhibitors, glycogen synthesis kinase 3 inhibitors, Akt inhibitors, and p38 MAPK inhibitors [6]. In a previous study, inhibition of p38 MAPK α and β by the selective inhibitor SB202190 was found to enhance 1,25D-induced myeloid differentiation [7]. Addition of SB202190 may result in removal of feedback inhibition for upstream regulators of MAPK pathways, such as ERK1/2, p38 MAPK γ/δ, and JNK1/2, resulting in activation of these pathways. Meanwhile, Akt inhibitors were reported to increase the percentage of myeloid differentiation marker CD11b [8] in AML M2-derived HL-60 cells [9]. Furthermore, Akt was found to regulate 1,25D-induced HL-60 cell differentiation through suppression of Raf/MEK/ERK MAPK signaling, with genetic silencing of Akt resulting in activation of this pathway [10]. We recently revealed that the Akt inhibitor triciribine (TCN) can efficiently induce the differentiation of NB4 APL cells and AML M2-derived HL-60 cells through activation of the ERK/MAPK pathway [11]. Taking these findings together, we hypothesized that combinations of TCN and p38 MAPK inhibitors may act synergistically to enhance ERK MAPK signaling and have efficient combinatorial effects on myeloid differentiation. In this study, we found that the combination of TCN with several p38 MAPK inhibitors had potent effects on myeloid differentiation.

## Materials and methods

### Cell culture

NB4, HL-60 [12, 13], THP-1 [14], and U937 [15] cells were cultured in RPMI medium containing 10% heat-inactivated fetal bovine serum under 5% $CO_2$ at 37˚C in a humidified atmosphere. The cell cultures were performed using an authenticated technique (https://www.sigmaaldrich.com/japan/ordering/technical-service/recipe-cc.html).

### Surface marker expression analysis by flow cytometry

For flow cytometry (FCM) analysis, approximately 2–3×$10^5$ NB4, HL-60, THP-1, and U937 cells were induced to differentiate at the concentrations, if not indicated, of 10 μM TCN and/or PD169316. ATRA (100 nM) was added to NB4 and HL-60 cells. Cells were incubated for 72h. After washing with phosphate-buffered saline, 30-μL aliquots of cell suspensions were protected from light and incubated with 1 μL of various antibodies (S1 Table) for 30 min at

room temperature. An isotype-matched PE-conjugated mouse IgG antibody (BioLegend, San Diego, CA) was used as a negative control. After the incubation, the cells were analyzed in an LSR Fortessa X-20 Flow Cytometer (BD Biosciences, San Jose, CA).

## Morphological analysis

NB4 and HL-60 cells in the logarithmic growth phase were seeded at $2 \times 10^5$ cells/mL and induced to differentiate with 10 µM TCN and/or 10 µM PD169316. The differentiated cells were collected for analysis at 72 h after the addition of reagents. Differentiation was evaluated by the cell morphologies after Wright–Giemsa staining. The nuclear and cytoplasmic areas were measured in individual cells and quantified using Image J software (https://imagej.nih.gov/ij/). The data were statistically analyzed and presented as scatterplots (violin plots) using Prism software (version 9.0; GraphPad Software Inc., La Jolla, CA).

## Western blotting

We used a Nuclear Extract and Cytosol Preparation Kit (Apro Science, Naruto, Tokushima, Japan) in accordance with the manufacturer's protocol. Appropriate amounts of 1× PhosSTOP Phosphatase Inhibitor Cocktail (Roche, Indianapolis, IN) and 1× Complete Protease Inhibitor Cocktail (Roche) were added, and the protein concentration was measured using a Pierce BCA Protein Assay Kit (Thermo Fisher, Waltham, MA). Aliquots of supernatants containing 10–20 µg of protein were separated by electrophoresis and immunoblotted. Images were captured using a LuminoGraph III(WSE-6300H-CS; ATTO, Tokyo, Japan). Total cellular extracts were also prepared and immunoblotted as described [11]. Signaling pathway molecules were examined using anti-phospho-p44/42 MAPK ERK1/2(Thr202/Tyr204), anti-phospho-p38 MAPK(T180/Y182), and anti-p44/42 MAPK mouse monoclonal antibodies (all from Cell Signaling Technology, Beverly, MA). An anti-p38 MAPK α/stress-activated protein kinase (SAPK) 2α mouse monoclonal antibody was obtained from BD Biosciences. A rabbit anti-β-actin monoclonal antibody (Cell Signaling Technology) was employed to confirm equal loading of protein.

## Microarray and mRNA expression analyses

For microarray analyses, total cellular RNA was isolated from control cells and NB4 cells at 72 h after addition of 10 µM TCN and/or 10 µM PD169316 using an RNA Mini Purification Kit (Qiagen, Miami, FL) in accordance with the manufacturer's protocol. The samples were subjected to microarray analyses using a Clariom™ D Assay (Filgen, Nagoya, Japan). Data were analyzed using Transcriptome Analysis Console software (Thermo Fisher Scientific, Waltham, MA). KEGG pathway analyses were conducted using the Database for Annotation, Visualization, and Integrated Discovery (DAVID) (https://david.ncifcrf.gov). The gene expression datasets have been deposited in the NCBI Gene Expression Omnibus (http://www.ncbi.nlm.nih.gov/geo/) under the GEO series accession number GSE235632. To prepare RNA for real-time PCR analyses, cells were seeded at a density of $2–3 \times 10^5$ cells/mL and treated with 10 µM TCN and/or 10 µM PD169316. The cells were harvested after 72 h or at specified times. cDNAs were prepared from the cells using a ReverTra Ace® qPCR RT Kit (Toyobo, Tokyo, Japan). Quantitative PCR was performed using THUNDERBIRD SYBR qPCRMix (Toyobo) in accordance with the manufacturer's protocol and an Opticon Mini Real-time PCR Instrument (Bio-Rad, Hercules, CA) as previously described [12]. The sequences of the primers are listed in S2 Table. The thermal cycling conditions for all genes were: 95˚C for 15 s; 40 cycles of 95˚C for 10 s, 60˚C for 45 s, and 72˚C for 3 min.

## Results

### Combination of TCN and p38 MAPK inhibitor PD169316 potently induces myeloid differentiation markers in NB4 and HL-60 cells

A number of kinase inhibitors have been reported to induce cell differentiation, including cyclin-dependent kinase inhibitors, glycogen synthesis kinase 3 inhibitors, Akt inhibitors, p38 MAPK inhibitors, Src family kinase inhibitors, Syk inhibitors, mTOR inhibitors, and HSP90 inhibitors [6]. In the present study, we first examined several of these kinase inhibitors, both alone and in combination. Flow cytometry analysis revealed that combinations of TCN with p38 MAPK inhibitors (PD169316, CAY10571, SB202190, SB203580) potently increased the expression of CD11b in NB4 cells (Fig 1A). Next, we examined the expression of other differentiation markers by real-time PCR using NB4 cells. We found that the expressions of not only *CD11b* (Fig 1B), but also *CD11c* (Fig 1C) and *CD14* (Fig 1D) were induced by these combinations, while the expression of *myeloperoxidase* (*MPO*) was efficiently down-regulated by these combinations (Fig 1E). We verified these phenomena using another myeloid cell line, AML M2-derived HL-60 cells, and found that similar inductions occurred (Fig 1B–1E). We further examined the lower (0, 0.1, 0.5, 1.0 and 5.0 µM) dosages of these two combinations. As a result, below 1.0 µM of TCN, there were almost no effect for the combination, in both NB4 (S1 Fig) and HL-60 cells (S2 Fig). However, we found that 5.0 µM of TCN had sufficient, and 10 µM of TCN had potent combination effect. These concentrations are clinically achievable, since specimens from primary AML blasts accumulated a median peak concentration of 4 µM (range of 2.1–7.5µM) [16]. To see the maximal effect, we choose 10 µM concentration for TCN and PD169316, for further study.

We further examined the effects of TCN and p38 MAPK inhibitor PD169316 on the expression of 18 differentiation markers (Fig 2, S1 Table) using flow cytometry. In NB4 cells (Fig 2A), we found that CD11b, CD11c, glycophorin A, and CD20 expressions were induced by these two agents. CD7 and CD14 expressions were induced by TCN, but were not affected by addition of PD169316. CD13 expression was induced by TCN, but this was suppressed by PD169316, and the combination of these two agents decreased its expression, in contrast to the expression in HL-60 cells (Fig 2B). When the effects of ATRA were examined, most of the marker expressions showed similar effects to those of TCN+PD169316, except for CD20, glycophorin A, and CD14. In HL-60 cells (Fig 2B), CD7, CD11b, CD11c, CD20, glycophorin A, and CD14 expressions were induced by these two agents, and the effects were mostly similar to those of ATRA. CD13 expression was induced by TCN, but was not affected by addition of PD169316. Collectively, in these myeloid cells, the combination of these two agents efficiently induced the expression of myelomonocytic (CD11b, CD11c) markers and some ectopic (erythroid glycophorin A, lymphoid CD7 and CD20) markers. As a comparison to myeloid NB4 and HL-60 cells, we examined the effect of these two agents in monocytic leukemia THP-1 cells [14] (Fig 2C) and human histiocytic lymphoma U937 cells [15] (Fig 2D). In THP-1 cells, CD2, CD3, CD5, CD11c, CD10, and CD19 expressions were induced, while MPO expression was suppressed, by these two agents. CD7, CD20, CD14, CD13, CD33, and CD41 expressions were induced by TCN, but were not affected by addition of PD169316. In U937 cells, only CD2 expression was induced by these two agents and almost no other markers were affected. These findings indicate that the effects of the combination of these two agents may differ between myeloid and monocytic lineages.

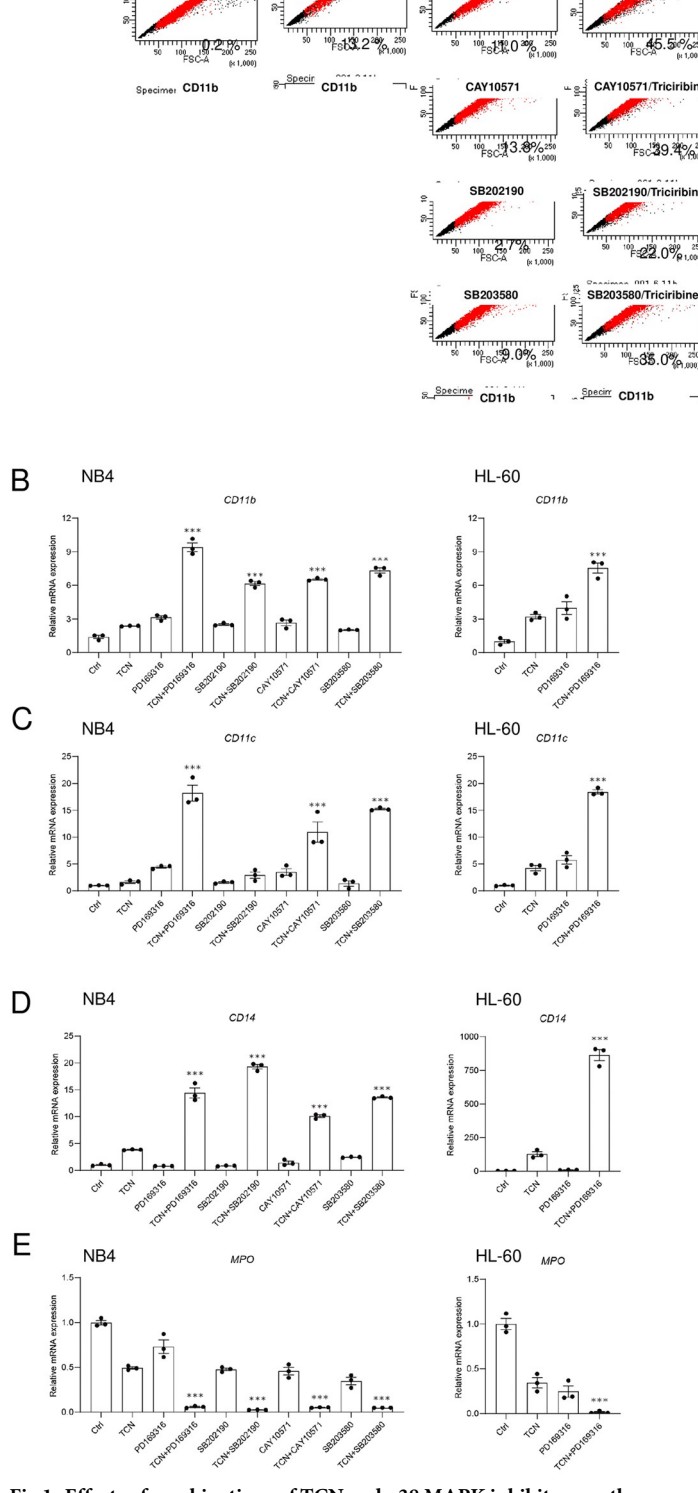

**Fig 1. Effects of combinations of TCN and p38 MAPK inhibitors on the expression of differentiation markers in NB4 cells.** (A) Percentages of CD11b-positive NB4 cells analyzed by flow cytometry after addition of the indicated kinase inhibitors. Representative histograms of at least three independent experiments are shown. (B–E) Effects of combinations of TCN and p38 MAPK inhibitors (PD169316, CAY10571, SB203580) on the differentiation of NB4 and HL-60 cells examined by real-time PCR analysis for several markers. The expressions of (B) CD11b, (C) CD11c, (D) CD14, and (E) MPO were examined. The data presented were obtained from three independent PCR amplifications,

and the reproducibility was confirmed using different batches of cDNA. Statistical significance was determined by two-way ANOVA followed by a Tukey multiple comparisons test (***p<0.001 vs. ctrl).

## Combination of TCN and PD169316 changes the morphology and potently decreases the nuclear-to-cytoplasmic ratio in NB4 and HL-60 cells

We examined the morphologies of NB4 and HL-60 cells at 72 h after the addition of TCN and/or PD169316. In NB4 cells (Fig 3A), the cytoplasmic area was enlarged with slight budding of the plasma membrane after addition of TCN. Addition of PD169316 also caused slight budding of the plasma membrane, while addition of both agents further enlarged the cytoplasmic region with obvious budding. For objective evaluation, we calculated the nuclear-to-cytoplasmic (N/C) ratio in NB4 cells. The data were statistically analyzed and presented as a violin plot. As shown in Fig 3B, the mean N/C ratio was 76.5% without inhibitors, compared with 66.2%–70.8% in the presence of TCN or PD169316 alone. With the combination of TCN and PD169316, the N/C ratio was significantly decreased to 53.4%, indicating further enhancement of differentiation by the combination of these two agents.

In HL-60 cells (Fig 3C), the cytoplasmic area was increased after addition of TCN or PD169316 alone, and the combination of these agents induced increased budding of the cells (arrows). Regarding the N/C ratio, addition of each agent alone significantly decreased the N/C ratio to 50.7% for TCN and 57.1% in PD169316, and the combination of these two agents significantly decreased the ratio to 41.4% (Fig 3D). Collectively, these findings indicate that the cell differentiations were not full myeloid differentiation [17, 18], but partial myelomonocytic differentiation. The combination of PD169316 and TCN resulted in budding of the cells (Fig 3A and 3C), as a characteristic feature of monocytic differentiation. Taken together with the induction of CD14 expression (Figs 1 and 2) in cells treated with PD169316 and TCN, the combination of these two kinase inhibitors may induce partial monocytic differentiation of the cells.

## Effect of TCN and PD169316 on signaling pathways in NB4 and HL-60 cells

In our recent study, we revealed that TCN can efficiently induce NB4 and HL-60 cell differentiation through activation of the ERK/MAPK pathway [11]. Therefore, we examined whether the combination of TCN and PD169316 can induce activation of ERK/MAPK. As shown in Fig 4, TCN and PD169316 each efficiently induced phosphorylation of ERK, while PD169316 inhibited phosphorylation of p38 MAPK, in both cell types. Although the enhanced effect of the combination had almost disappeared at 24 h after addition of the agents, a marginal increase in ERK phosphorylation was observed in both cell types at 8 h, suggesting that ERK may be involved in the effects of the combination of these agents. In contrast, TCN and PD169316 had almost no effect on p38 MAPK phosphorylation. These findings suggest that ERK activation may play a role in the combined effect of TCN and PD169316.

## Gene expression profiling to reveal the effect of TCN and PD169316

To clarify the molecular mechanisms for the effects of TCN and PD169316 on NB4 cell differentiation, we performed microarray analyses using NB4 cells. NB4 cells were harvested at 72 h after addition of TCN and/or PD169316. We employed part of the data from our previous study investigating the effects of TCN on NB4 cells [11], for comparison with the effects of PD169316. When defined as >3-fold changes compared with the control, TCN up-regulated

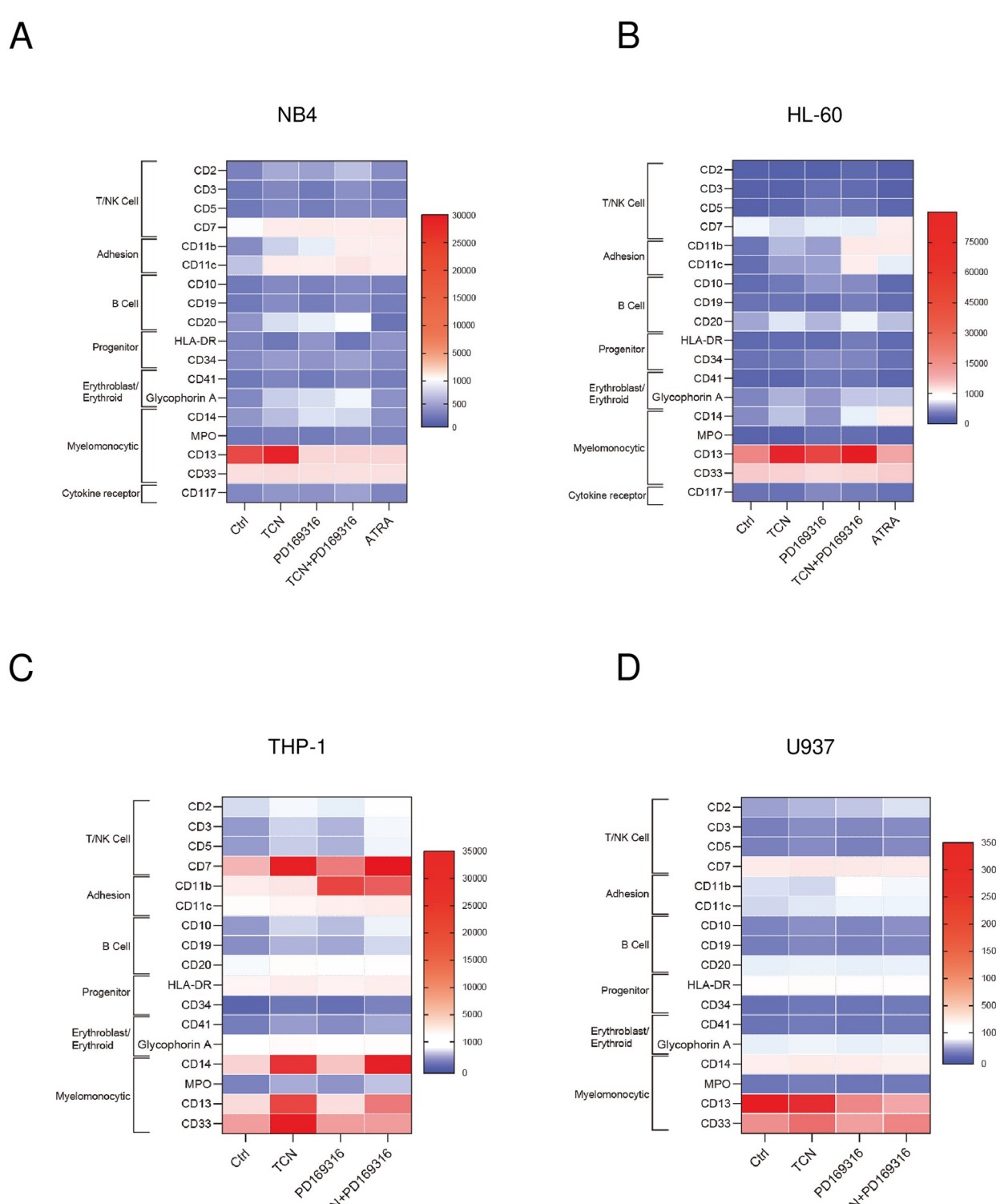

**Fig 2. Flow cytometry analyses of differentiation markers in NB4, HL-60, THP-1, and U937 cells.** (A–D) Effects of combinations of TCN and PD169316 on the differentiation of (A) NB4, (B) HL-60, (C) THP-1, and (D) U937 cells examined by flow cytometry. In NB4 (A) and HL-60 (B) cells, ATRA was added for a comparison. The expressions of 18 differentiation markers were examined. The heat maps show the expression levels of the leukocyte cell surface markers in terms of the geometric mean fluorescence intensity. The data presented were obtained from three independent flow cytometry analyses, and each box shows the mean geometric mean fluorescence intensity of three analyses.

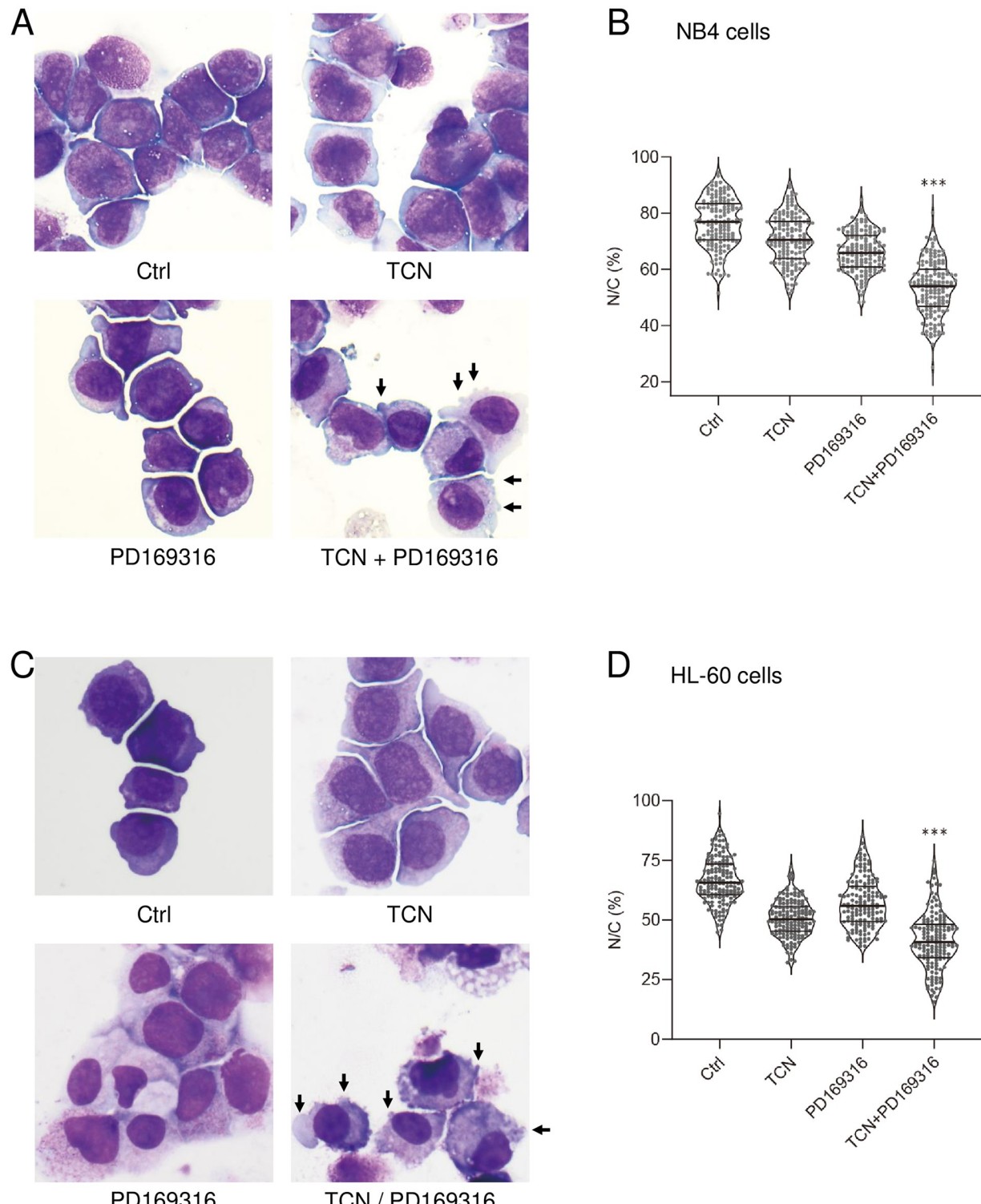

**Fig 3. Morphologic analyses of NB4 cells and HL-60 cells in the presence or absence of TCN and/or PD169316.** (A) NB4 cells were collected and subjected to Wright–Giemsa staining. Budding of the cells is indicated by arrows. The magnification of the images shown in the panels is ×400. (B) Violin plot analysis of the N/C ratio percentages in individual NB4 cells. The N/C ratio was calculated from the cytoplasm and nuclear areas using Image J software. Significance was analyzed (***p<0.001). The mean values and 25 percentiles are shown as thick and thin lines, respectively. (C) HL-60 cells were collected and subjected to Wright–Giemsa staining. Budding of the cells is indicated by arrows. The magnification of the images shown in the panels is ×400. (D) Violin plot analysis of the N/C ratio percentages in individual HL-60 cells. The data were analyzed as described for (B).

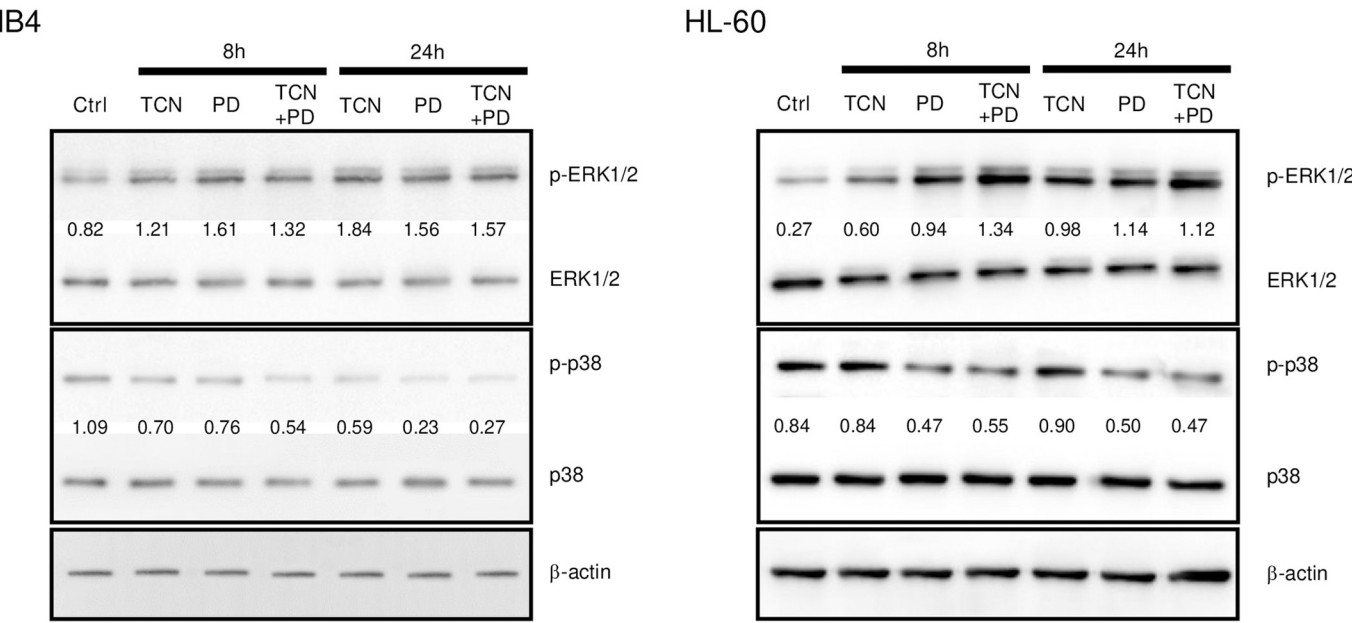

**Fig 4. Effects of TCN and PD169316 on signaling pathways in NB4 and HL-60 cells.** Cells were cultured in the presence or absence of TCN and/or PD169316 for the indicated periods, and analyzed for ERK and p38 MAPK activity by western blotting. The membranes were probed with an anti-phospho-ERK rabbit polyclonal antibody or anti-phospho-p38 MAPK mouse monoclonal antibody. The membranes were stripped and re-probed with an anti-total-ERK mouse monoclonal antibody or anti-total-p38 MAPK mouse monoclonal antibody to verify equal protein loading. The indicated numbers are the relative densities calculated by Image J 1.54 software, and were based on the densities of the bands for phospho-ERK or phospho-p38 MAPK divided by total-ERK or total-p38 MAPK, respectively.

348 genes and down-regulated 83 genes (Fig 5A) [11], while PD169316 up-regulated 172 genes and down-regulated 148 genes (Fig 5B). Furthermore, when defined as >5-fold changes compared with the control, the combination of TCN plus PD169316 up-regulated 510 genes and down-regulated 208 genes (Fig 5C). We conducted pathway analyses for the 510 up-regulated genes using DAVID software, and revealed that "viral protein interaction with cytokine and cytokine receptor" and "cytokine-cytokine receptor interaction" were enriched with high significance (Fig 5D). Lists of the genes enriched for "viral protein interaction with cytokine and cytokine receptor" and "cytokine-cytokine receptor interaction" in the analysis are shown in S3 and S4 Tables, respectively. We then focused on these pathways. Pathway analyses were conducted using the KEGG pathway database and DAVID software, and the results for "viral protein interaction with cytokine and cytokine receptor" and "cytokine-cytokine receptor interaction," are shown in S3 and S4 Figs, respectively. The red stars indicate the genes whose expressions were induced by >5-fold by TCN and PD169316, compared with the control.

Among the genes listed in S3 and S4 Tables, we selected 15 genes (shown in bold) and verified them by real-time PCR. In NB4 cells (Fig 6), chemokines like *CCL1*, *CCL2*, *CCL3*, *CCL5*, *CSF1*, *CXCL8* as well as cytokines and receptors like *IL-10*, *IL-10RA*, *IL-10RB*, *IL-1β*, *IL- RB*, and *TNSF10* were significantly upregulated by TCN and PD169316 for cell differentiation. We conducted the same experiments in HL-60 cells (Fig 6) and found that similar genes, namely *CCL1*, *CCL2*, *CCL3*, *CCL5*, *CSF1*, *CSF1R*, *CSF2RA*, *CXCL8*, *IL-10*, *IL-10RA*, *IL-10RB*, *IL-1β*, *TNFSF10*, and *TNFSF15*, were significantly induced by TCN and PD169316 for cell differentiation. Collectively, *CCL1*, *CCL2*, *CCL3*, *CCL5*, *CSF1*, *CXCL8*, *IL-10*, *IL-10RA*, *IL-10RB*, *IL-1β*, and *TNFSF10* were upregulated in both NB4 and HL-60 cells during TCN and PD169316-induced differentiation.

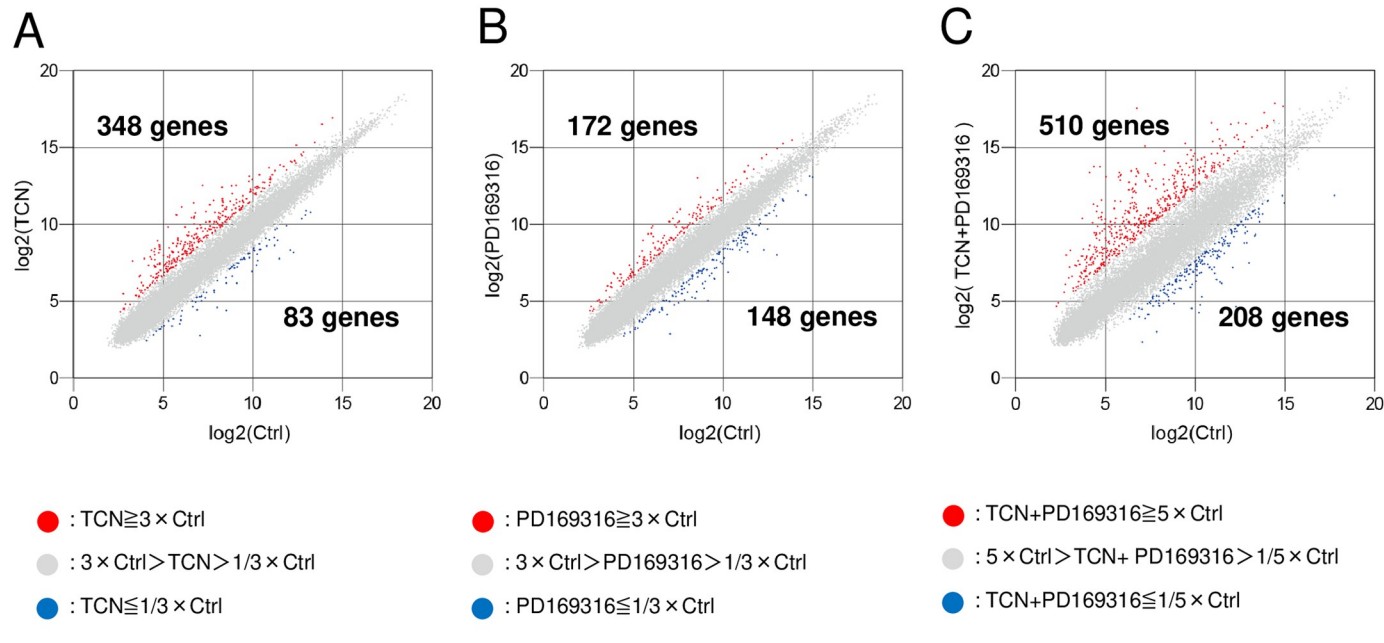

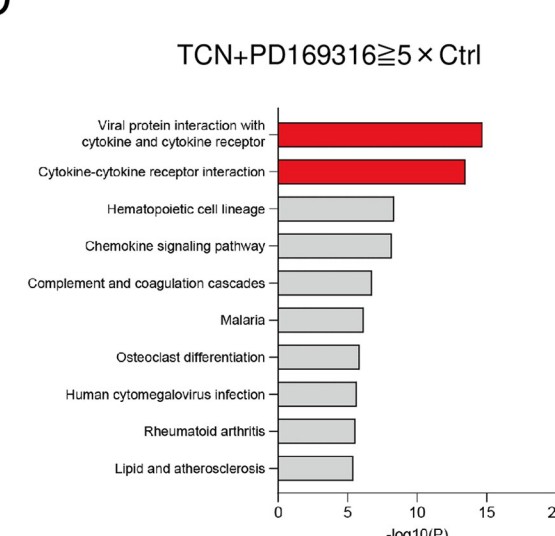

**Fig 5. Results of the microarray analyses.** (A) Gene expression profiles of NB4 cells treated with TCN versus control cells. The results of this comparison were already presented in a previous study [11]. (B) Gene expression profile of NB4 cells treated with PD169316 versus control cells. (C) Gene expression profile of NB4 cells treated TCN+PD169316 versus control cells. (D) Results of the pathway analysis using DAVID software (https://david.ncifcrf.gov/home.jsp) for 510 genes. The X-axis shows the number of the negative log base 10 p-values. Higher numbers indicate higher significance.

## Discussion

Sampath et al. [16] reported that in advanced hematological malignancies, TCN phosphate monohydrate therapy was well tolerated. Therefore, TCN combinations are worthy of exploration in clinical trials. TCN is a highly specific Akt inhibitor, and inhibition of Akt was shown to

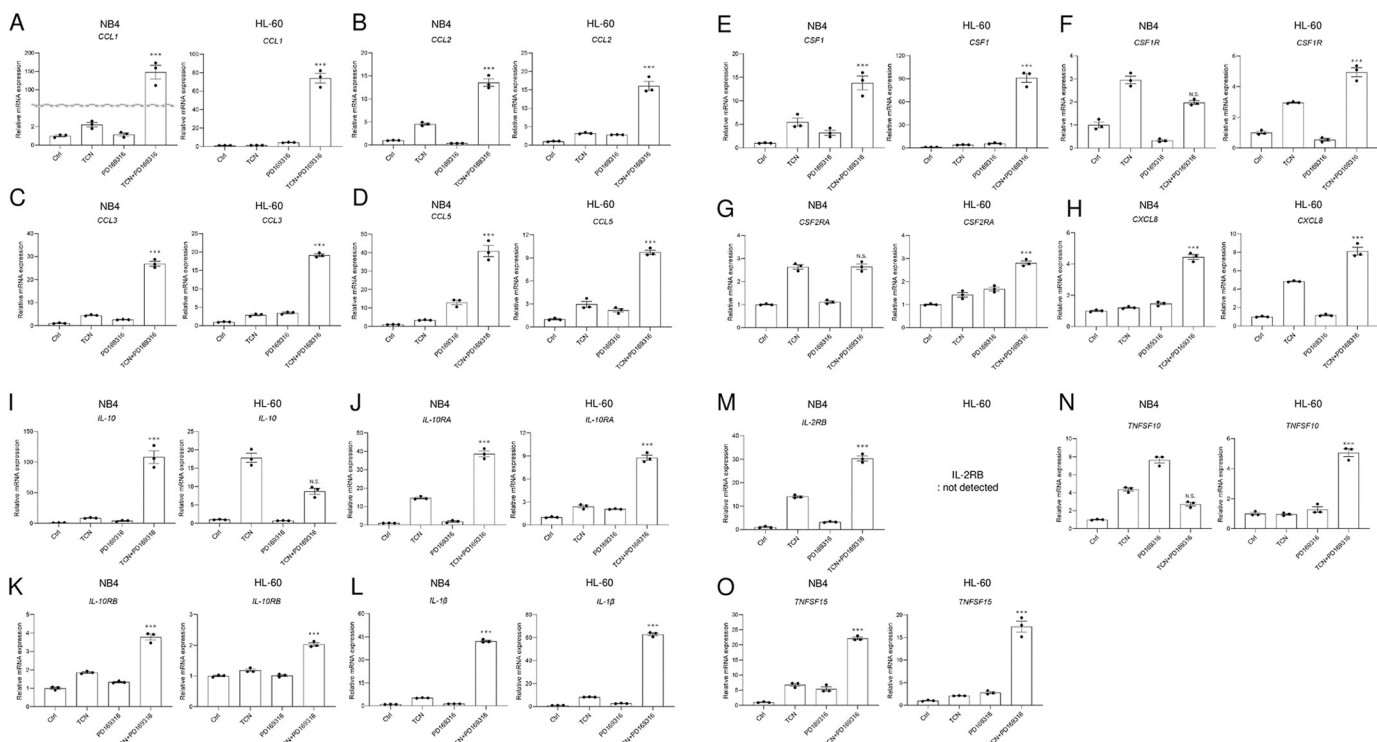

**Fig 6. Validation of the microarray data for selected genes in NB4 and HL-60 cells.** (A–O) Total RNAs extracted from NB4 cells cultured in the presence or absence of TCN and PD169316 for 72 h were analyzed by real-time PCR with specific primers for (A) *CCL1*, (B) *CCL2*, (C) *CCL3*, (D) *CCL5*, (E) *CSF1*, (F) *CSF1R*, (G) *CSF2RA*, (H) *CXCL8*, (I) *IL-10*, (J) *IL-10RA*, (K) *IL-10RB*, (L) *IL-1β*, (M) *IL-2RB*, (N) *TNFSF10*, and (O) *TNFSF15*. The data presented were obtained from three independent PCR amplifications, and the reproducibility was confirmed using different batches of cDNA. Statistical significance was determined by two-way ANOVA followed by a Tukey multiple comparisons test (***p<0.001 vs. ctrl).

activate the Raf/MEK/ERK pathway [10]. Because the Raf/MEK/ERK pathway is considered to control the balance between differentiation and expansion of hematopoietic progenitors [19], activation of this pathway by Akt inhibitors may play a pivotal role in the mechanisms of myeloid differentiation. Other p38 MAPK inhibitors, SB203580 or SB202190, alone could induce monocytic, but not macrophage or granulocytic, differentiation of some myeloid leukemia cells, HL-60, HT93, and ML-1, but did not induce differentiation of monocytoid leukemia U937 and THP-1 cells [20, 21]. These findings are consistent with the present findings showing that synergistic differentiation by TCN and PD169316, a p38 MAPK inhibitor, specifically affected myeloid cells (NB4, HL-60), but did not affect monocytic (THP-1) or histiocytic (U937) cells. The two p38 MAPK inhibitors SB202190 and SB203580 increased ERK activity, independent of p38 MAPK activity, in the myeloid leukemia cell line HL-60 and erythroleukemic cell line TF-1 [22, 23]. We speculate that insufficient reactions by p38 MAPK inhibitors in monocytic cells may result in the lack of differentiation effect by TCN and p38 MAPK inhibitors.

The Raf/MEK/ERK pathway plays an important role in hematopoietic stem cells. Geest et al. [19] reported that tight control of MEK/ERK activation is essential for regulating survival and cytokine production in CD34-positive neutrophil progenitors. They demonstrated that conditional activation of MEK1 led to a dramatic increase in the expression of mRNAs encoding a large number of chemokines and growth factors similar to the genes observed in our microarray analyses. We found that chemokines (CCL1, CCL2, CCL3, CCL5, CXCL8) as well as cytokines and their receptors (IL-10, IL-10RA, IL-10RB, IL-1β, TNSF15) were significantly

induced by TCN and PD169316 for cell differentiation. It was reported that induction of the chemokines we identified (CCL1, CCL2, CCL3, CCL5, CXCL8) plays a role in the suppression of hematopoiesis [24], consistent with the features of differentiation. We further revealed that the IL-10 pathway was induced by these agents. IL-10 is a growth and differentiation factor for B-cells [25]. Moreover, IL-10 can promote cytotoxic T-cell responses [25]. Consistent with this, we demonstrated that the combination of TCN and PD169316 efficiently induced the T cell marker CD7 and B cell marker CD20. Moreover, we found that the tumor necrosis factor superfamily (TNFSF) 10 and 15 genes, both of which play a role in T cell function [26], were significantly induced by TCN and PD169316, and may have a role in the T cell development induced by the addition of these agents.

The results of clinical trials on p38 MAPK inhibitors (Ralimetinib [LY2228820], Dilmapimod [SB-681323], Neflamapimod [VX-745]) for advanced cancer [27], neuropathic pain following nerve injury [28], and Alzheimer's disease [29] have been reported. In these studies, the p38 MAPK inhibitors were well tolerated with acceptable toxicities. TCN was also reported to exhibit limited toxicity in various malignancies, including hematological malignancies [16], solid tumors [30], metastatic breast cancer [31], and squamous carcinoma of the cervix [32]. Therefore, basic and clinical research, including combination trials of TCN and p38 MAPK inhibitors for APL and non-APL leukemia, are worthy of exploration. The differentiation mechanisms and signaling routes of TCN, as well as PD169316, remain to be elucidated in future studies.

## Supporting information

**S1 Fig. Effects of combinations of TCN and PD169316 on the differentiation of NB4 cells examined by flow cytometry analysis.** The concentration of each reagent was 0, 0.1, 0.5, 1.0, 5.0 and 10 μM. The bar graphs show the expression levels of the CD11b surface markers in terms of the geometric mean fluorescence intensity. Data are presented as the mean±standard error (SE) (n = 3/group).
(PPTX)

**S2 Fig. Effects of combinations of TCN and PD169316 on the differentiation of HL-60 cells examined by flow cytometry analysis.** Similar to S1 Fig.
(PPTX)

**S3 Fig. Results of the pathway analysis using the KEGG pathway database and DAVID software.** The 510 genes in "viral protein interaction with cytokine and cytokine receptor" were evaluated. The red stars indicate genes with expressions induced by >5-fold by TCN and PD169316 compared with the control.
(PPTX)

**S4 Fig. Results of the pathway analysis using the KEGG pathway database and DAVID software.** The 510 genes in "cytokine-cytokine receptor interaction" were evaluated. The red stars indicate genes with expressions induced by >5-fold by TCN and PD169316 compared with the control.
(PPTX)

**S1 Table. List of antibodies used for the flow cytometry analyses.**
(XLSX)

**S2 Table. Sequences of primers used for the real-time quantitative PCR analyses.**
(XLSX)

**S3 Table. List of genes for "viral protein interaction with cytokine and cytokine receptor" in the pathway analysis using the KEGG pathway database and DAVID software.** Bold text indicates the genes selected and verified by real-time PCR.
(XLSX)

**S4 Table. List of genes for "cytokine-cytokine receptor interaction" in the pathway analysis using the KEGG pathway database and DAVID software.** Bold text indicates the genes selected and verified by real-time PCR.
(XLSX)

**S1 Raw images.**
(PDF)

## Acknowledgments

The authors thank Alison Sherwin, PhD, from Edanz (https://jp.edanz.com/ac) for editing a draft of this manuscript.

## Author Contributions

**Conceptualization:** Shinichiro Takahashi.

**Data curation:** Susumu Suzuki, Shinichiro Takahashi.

**Formal analysis:** Susumu Suzuki, Shinichiro Takahashi.

**Funding acquisition:** Shinichiro Takahashi.

**Investigation:** Yuri Sato-Nagaoka, Susumu Suzuki, Souma Suzuki.

**Project administration:** Shinichiro Takahashi.

**Resources:** Shinichiro Takahashi.

**Writing – original draft:** Shinichiro Takahashi.

**Writing – review & editing:** Shinichiro Takahashi.

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
