## [Decision Letter · Decision Letter 0]

21 Aug 2024

PONE-D-24-25935Combination of triciribine and p38 MAPK inhibitor PD169316 enhances the differentiation effect on myeloid leukemia cellsPLOS ONE

Dear Dr. Takahashi,

Thank you for submitting your manuscript to PLOS ONE. After careful consideration, we feel that it has merit but does not fully meet PLOS ONE’s publication criteria as it currently stands. Therefore, we invite you to submit a revised version of the manuscript that addresses the points raised during the review process.

Your manuscripts have been reviewed by two experts in the field of cancer and gene transcription. While they found your results promising, they also identified some important issues, including (but are not limited to) cell line differences, some poor-quality data, and questions about the clinical relevance of the conditions used in your study. All these issues must be addressed by substantially revising the current manuscript. Please ensure that your decision is justified on PLOS ONE’s publication criteria and not, for example, on novelty or perceived impact.

We look forward to receiving your revised manuscript.

Kind regards,

Jinsong Zhang

Academic Editor

PLOS ONE

Journal Requirements:

2. In your Methods section, please report the source of cell lines used for your study.

https://journals.plos.org/plosone/article?id=10.1371%2Fjournal.pone.0303428

https://karger.com/aha/article-abstract/145/2/113/821321/Kinase-Inhibitors-and-Interferons-as-Other-Myeloid?redirectedFrom=fulltext

https://linkinghub.elsevier.com/retrieve/pii/S0006291X24000767

In your revision ensure you cite all your sources (including your own works), and quote or rephrase any duplicated text outside the methods section. Further consideration is dependent on these concerns being addressed.

4. Thank you for stating the following financial disclosure: Grant-in-Aid for Scientific Research (21K07346) from the Ministry of Education, Science and Culture, Japan, to ST.

Kyowa-Kirin Research Support (Kyowa-Kirin Co. Ltd.), Japan, to ST.

Daiichi-Sankyo Research Support (Daiichi-Sankyo Inc.), Japan, to ST. 

Reviewers' comments:

Reviewer's Responses to Questions

**Comments to the Author**

1. Is the manuscript technically sound, and do the data support the conclusions?

Reviewer #1: No

Reviewer #2: Yes

2. Has the statistical analysis been performed appropriately and rigorously? 

Reviewer #1: Yes

Reviewer #2: I Don't Know

3. Have the authors made all data underlying the findings in their manuscript fully available?

Reviewer #1: Yes

Reviewer #2: Yes

4. Is the manuscript presented in an intelligible fashion and written in standard English?

Reviewer #1: Yes

Reviewer #2: Yes

5. Review Comments to the Author

Reviewer #1: In this manuscript Combination of triciribine and p38 MAPK inhibitor PD169316 enhances the differentiation effect on myeloid leukemia cells, Sato-Nagaoka et al used two cell lines NB4 and HL-60 to study the effects of TCN and PD169316 on leukemic cell differentiation. They showed that the combination of two agents had an enhanced effect on cell differentiation. They further investigated the mechanism underlying the observed effect and found that multiple pathways and genes were involved. Overall, this is an interesting study with intriguing observations.

However, there are some concerns which need to be addressed.

1.Both TCN and PD169316 were used at 10uM. How did the authors choose this dosage? Do lower doses have an effect on cell differentiation? Is this dose clinically relevant if a clinical trial is to be conducted. Authors need to provide more information on their choice of dosage.

2.Figure 2. CD13 expression was increased after TCN+PD169316 in HL-60 cells, but CD13 was decreased after TCN+PD169316 treatment based on the heatmap shown. Please explain this result in the text.

3.Figure 4. p-ERK1/2 Western blot in NB4 cells was overexposed and saturated. The difference between bands was not obvious. Please repeat the experiemnts and have high quality publishable band images available. The b-Actin loading control bands were not of the publication quality for NB4 cells in Figure 4.

Reviewer #2: This study evaluates the effects of the p38 MAPK inhibitor PD169316 in enhancing the differentiation effects of the Akt inhibitor triciribine triciribine (TCN) on myeloid leukemia cell differentiation. In a previous study the authors showed that PD169316, SB203580, SB202190 (p38 MAPK inhibitors), and TCN potently increased the expression of CD11b in the NB4 acute promyelocytic leukemia cell line (PLoS One. 2024 May 14;19(5):e0303428). While that study focuses on TCN, this study focuses on the combined effects of PD169316 and TCN. The methods used are very similar. The results show enhancement of combination of TCN and PD169316 potently induces myeloid differentiation markers in NB4 and HL-60 cells, changes the morphology and decreases the nuclear-to-cytoplasmic ratio, induced phosphorylation of ERK, and induces a panel of chemokines as well as cytokines and their receptors. The data is of high quality. However, the effects are shown in M3 NB4 cells which contains two t(15;17) chromosomes and other cytogenetic abnormalities, and the M2 HL-60 cells, both of which can be differentiated into granulocytic and mononuclear lineage. The effects were absent in M5 acute monoblastic and monocytic leukemia cell lines (THP-1 and U937). In the future, if the authors could demonstrate the key effects (e.g. myeloid differentiation markers and nuclear-to-cytoplasmic ratio) in a couple more AML cell lines, it could further validate the effects of TCN and PD169316.

6. PLOS authors have the option to publish the peer review history of their article (what does this mean?). If published, this will include your full peer review and any attached files.

Reviewer #1: No

Reviewer #2: No

---

## [Author Response · Author response to Decision Letter 0]

18 Sep 2024

Reviewer #1: In this manuscript Combination of triciribine and p38 MAPK inhibitor PD169316 enhances the differentiation effect on myeloid leukemia cells, Sato-Nagaoka et al used two cell lines NB4 and HL-60 to study the effects of TCN and PD169316 on leukemic cell differentiation. They showed that the combination of two agents had an enhanced effect on cell differentiation. They further investigated the mechanism underlying the observed effect and found that multiple pathways and genes were involved. Overall, this is an interesting study with intriguing observations.

However, there are some concerns which need to be addressed.

1.Both TCN and PD169316 were used at 10uM. How did the authors choose this dosage? Do lower doses have an effect on cell differentiation? Is this dose clinically relevant if a clinical trial is to be conducted. Authors need to provide more information on their choice of dosage.

To address this issue, we examined the expression levels of differentiation marker CD11b, in lower doses of TCN and PD169316. As a result, we observed that above 5 µM of TCN and PD169316 had sufficient induction activity both in NB4 and HL-60 cells (Fig S1, S2). These are added as follows (page 9, line 167).

We further examined the lower (0, 0.1, 0.5, 1.0 and 5.0 µM) dosages of these two combinations. As a result, below 1.0 µM of TCN, there were almost no effect for the combination, in both NB4 (Fig. S1) and HL-60 cells (Fig. S2). However, we found that 5.0 µM of TCN had sufficient, and 10 µM of TCN had potent combination effect. These concentrations are clinically achievable, since specimens from primary AML blasts accumulated a median peak concentration of 4 µM (range of 2.1-7.5µM) [14]. To see the maximal effect, we choose 10 µM concentration for TCN and PD169316, for further study. 

2.Figure 2. CD13 expression was increased after TCN+PD169316 in HL-60 cells, but CD13 was decreased after TCN+PD169316 treatment based on the heatmap shown. Please explain this result in the text.

To follow this comment, we added following sentence in the text (line 192). 

CD13 expression was induced by TCN, but this was suppressed by PD169316, and the combination of these two agents decreased its expression, in contrast to the expression in HL-60 cells (Fig. 2B).

3.Figure 4. p-ERK1/2 Western blot in NB4 cells was overexposed and saturated. The difference between bands was not obvious. Please repeat the experiemnts and have high quality publishable band images available. The b-Actin loading control bands were not of the publication quality for NB4 cells in Figure 4.

To follow this comment, we repeated experiments for NB4 cells and obtained better quality, not saturated, blot for revised Figure 4A.

Following these comments, we feel the improvement of our paper. We appreciate very much for these comments.

Reviewer #2: This study evaluates the effects of the p38 MAPK inhibitor PD169316 in enhancing the differentiation effects of the Akt inhibitor triciribine triciribine (TCN) on myeloid leukemia cell differentiation. In a previous study the authors showed that PD169316, SB203580, SB202190 (p38 MAPK inhibitors), and TCN potently increased the expression of CD11b in the NB4 acute promyelocytic leukemia cell line (PLoS One. 2024 May 14;19(5):e0303428). While that study focuses on TCN, this study focuses on the combined effects of PD169316 and TCN. The methods used are very similar. The results show enhancement of combination of TCN and PD169316 potently induces myeloid differentiation markers in NB4 and HL-60 cells, changes the morphology and decreases the nuclear-to-cytoplasmic ratio, induced phosphorylation of ERK, and induces a panel of chemokines as well as cytokines and their receptors. The data is of high quality. However, the effects are shown in M3 NB4 cells which contains two t(15;17) chromosomes and other cytogenetic abnormalities, and the M2 HL-60 cells, both of which can be differentiated into granulocytic and mononuclear lineage. The effects were absent in M5 acute monoblastic and monocytic leukemia cell lines (THP-1 and U937). In the future, if the authors could demonstrate the key effects (e.g. myeloid differentiation markers and nuclear-to-cytoplasmic ratio) in a couple more AML cell lines, it could further validate the effects of TCN and PD169316.

We agree with this suggestion. Indeed, we are now investigating this combination in many AML cells, because we also think it is important and to increase the relevance of this series of research. However, it would be grateful if you allow as to move forward at this time, and clarify this as a next project. Thank you so much for this constructive comment.

We confirmed. Thank you.

2. In your Methods section, please report the source of cell lines used for your study.

We cited the source of cell lines used for this study.

https://journals.plos.org/plosone/article?id=10.1371%2Fjournal.pone.0303428

https://karger.com/aha/article-abstract/145/2/113/821321/Kinase-Inhibitors-and-Interferons-as-Other-Myeloid?redirectedFrom=fulltext

https://linkinghub.elsevier.com/retrieve/pii/S0006291X24000767

In your revision ensure you cite all your sources (including your own works), and quote or rephrase any duplicated text outside the methods section. Further consideration is dependent on these concerns being addressed.

We ensured to cite all of our sources, and all of these are from this corresponding author’s paper, as ref [6], [11], [13]. Furthermore, we rephrased related sentences.

4. Thank you for stating the following financial disclosure: Grant-in-Aid for Scientific Research (21K07346) from the Ministry of Education, Science and Culture, Japan, to ST.

Kyowa-Kirin Research Support (Kyowa-Kirin Co. Ltd.), Japan, to ST.

Daiichi-Sankyo Research Support (Daiichi-Sankyo Inc.), Japan, to ST. 

We added, Role of Funder statement: The funders had no role in study design, data collection and analysis, decision to publish, or preparation of the manuscript, in the cover letter.

We submitted our blot/gel image data as supporting information and this was noted in our cover letter. Thank you.

---

## [Decision Letter · Decision Letter 1]

7 Oct 2024

Combination of triciribine and p38 MAPK inhibitor PD169316 enhances the differentiation effect on myeloid leukemia cells

PONE-D-24-25935R1

Dear Dr. Takahashi,

We’re pleased to inform you that your manuscript has been judged scientifically suitable for publication and will be formally accepted for publication once it meets all outstanding technical requirements.

Kind regards,

Jinsong Zhang

Academic Editor

PLOS ONE

Additional Editor Comments (optional):

Reviewers' comments:

Reviewer's Responses to Questions

**Comments to the Author**

1. If the authors have adequately addressed your comments raised in a previous round of review and you feel that this manuscript is now acceptable for publication, you may indicate that here to bypass the “Comments to the Author” section, enter your conflict of interest statement in the “Confidential to Editor” section, and submit your "Accept" recommendation.

Reviewer #1: All comments have been addressed

2. Is the manuscript technically sound, and do the data support the conclusions?

Reviewer #1: Yes

3. Has the statistical analysis been performed appropriately and rigorously? 

Reviewer #1: Yes

4. Have the authors made all data underlying the findings in their manuscript fully available?

Reviewer #1: Yes

5. Is the manuscript presented in an intelligible fashion and written in standard English?

Reviewer #1: Yes

6. Review Comments to the Author

Reviewer #1: The authors have answered my questions and addressed my concerns. The manuscript could be considered for publication after editorial review.

7. PLOS authors have the option to publish the peer review history of their article (what does this mean?). If published, this will include your full peer review and any attached files.

Reviewer #1: No

---

## [Editor Report · Acceptance letter]

10 Oct 2024

PONE-D-24-25935R1 

PLOS ONE

Dear Dr. Takahashi, 

I'm pleased to inform you that your manuscript has been deemed suitable for publication in PLOS ONE. Congratulations! Your manuscript is now being handed over to our production team.

Kind regards, 

on behalf of

Dr. Jinsong Zhang 

Academic Editor

PLOS ONE